# Online black-box adaptation to generalized target-shift

## Abstract

We explore the use of test-time pseudo-labels for online label-shift adaptation when deploying black-box models. Specifically, we focus on settings where predictive models are deployed in new locations (leading to *conditional-shift*), such that these locations are also associated with differently skewed target distributions (label-shift), a combination more broadly referred to as generalized target-shift. Adapting Bayesian tools, we illustrate empirically that online estimates of label-shift using pseudo-labels can often be beneficial in such settings, even with the conditional-shift associated with different deployment locations, when hyper-parameters are learned on validation sets. We illustrate the potential of this approach on three synthetic and two realistic datasets comprising both classification and regression problems.

## 1 Introduction

We consider a practical setting where we have black-box access to a predictive model which we are interested in deploying in different places with skewed label distributions. Such situations can arise, for example, in healthcare settings, where there are often differing rates of disease-incidence (Vos et al., 2020) accompanied by distributional shift in input features at different deployment locations, for example with diagnostic radiology Cohen et al. (2021). In notation, for input variable $x$ and target variable $y$, we have that $\mathbb{P}^{\text{train}}(x \mid y) \neq \mathbb{P}^{\text{test}}(x \mid y)$ (conditional-shift) and $\mathbb{P}^{\text{train}}(y) \neq \mathbb{P}^{\text{test}}(y)$ (label-shift), a combination broadly referred to in the domain adaptation literature as *generalized target-shift* (GETARS) (Zhang et al., 2013).

**Contextual information can be useful** The prevailing notion in current considerations of *out-of-distribution* (OOD) generalization tends to promote agnosticism to background or 'non-causal' information, since these are deemed to be potentially biasing. While this seems fairly sensible and certainly the reason behind some failure-modes, sometimes background information can serve a purpose. Consider a poorly-lit scene with a highly-occluded spotted wild cat; this could be either a leopard or a jaguar. However, leopards and jaguars tend to have geographical biases – leopards are found in Africa and Asia while jaguars are found in Central and South America. In such situations, contextual information can help resolve confusion, not only for an AI system but perhaps also for the human expert who labelled the training data (Terry et al., 2020). This can especially be leveraged in settings where contextual factors like deployment location skews the label-distribution. To use the leopard/jaguar example – even if we do not know the label-distribution in a new deployment, after observing a number of leopards and no jaguars, we would find an occluded spotted wild cat less ambiguous.

**Real-life deployments can come with practical constraints** Consider a cloud-based AI service provider servicing clients operating in real-time and numbering in the hundreds of thousands. Such providers typically work with private training data and a proprietary model they are unwilling to share publicly. While context-based adaptation of model parameters can provide personalized results, as discussed above, it is often not possible for a provider to create as many copies of a model, especially as our models grow ever larger. Placing copies of the model on-chip for every client would be expensive and would be associated with specific assumptions about access to specialized hardware. In this paper, we adopt considerations of such practically-motivated test-time restrictions in the context of realistic distributional shifts that are likely to arise in multiple deployments. Under these settings, we show that it is often possible to improve upon base performance by cheap output adjustments.

**Problem setting**   We assume a test-time setting with the following characteristics.

1. *Black-box model, private training data:* We assume we cannot fine-tune the parameters of our underlying predictive model, nor are we allowed access to examples of training data. This situation can arise in practical settings where we use cloud-based, proprietary services trained on large, private datasets (for example, Google's Vision APIs).

2. *Online predictions:* We operate in an online setting, where we wish to make predictions with streaming test data in different deployment locations, without access to a representative sample of test data in each of these locations upfront.

**Contributions**   We explore the possibility of adapting the outputs of such black-box models online to label-shifts, when picking hyper-parameters with a validation set of data also exhibiting GeTarS. Using Bayesian tools to estimate label-shifts with pseudo labels in different deployment locations, we evaluate the applicability of this approach using three proof-of-concept synthetic tasks and two challenging real-world datasets comprising both classification and regression. Our results suggest that such approaches might have the potential to provide a cheap and efficient way to achieve improvements over baseline performance.

## 2   Approach

We consider classification or regression tasks, with input variable $x$ and target variable $y$, such that the input data come from a set of different sources $s$ (deployment locations) at test-time. These sources are associated with particular distributions of the target variable, $\mathbb{P}(y \mid s)$.

**Predictive distribution**   The underlying predictive model comprises a feature extractor $f_\theta(x)$, which is either a continuous output for regression, or is combined with a linear layer $w$ to produce logits for classification. This provides us with $\mathbb{P}(y \mid x)$. Since we want to incorporate information from $s$, we want to model the predictive distribution $\mathbb{P}(y \mid x, s)$. Assuming that an input $x$ and a source $s$ are independent given the target $y$, as in Mac Aodha et al. (2019); Chu et al. (2019), we can derive (see Appendix A.1)

$$\mathbb{P}(y \mid x, s) \propto \frac{\mathbb{P}(y \mid x)\,\mathbb{P}(y \mid s)}{\mathbb{P}(y)}, \tag{1}$$

where $\mathbb{P}(y)$ is estimated from the training data. In the applications described in the next sub-sections, we will use normalized class frequencies in the training set for representing $\mathbb{P}(y)$ in classification tasks and model $\mathbb{P}(y)$ as a Gaussian with the training mean and variance for regression tasks. Since such summary statistics of training labels do not expose specific $(x, y)$ examples, providers should not be opposed to sharing this information if the adaptation is to be performed at the client-end. When training labels are known to be approximately uniform, one can drop the $\mathbb{P}(y)$ term, yielding a product-of-experts model (Hinton, 2002).

**Posterior predictive distribution at $s$**   We represent $\mathbb{P}(y \mid s)$ using the posterior predictive distribution (Gelman et al., 2013) after seeing $n$ test examples in $s$ as

$$\mathbb{P}_n(y \mid s) = \int_\phi \mathbb{P}(y \mid \phi)\,\mathbb{P}_n(\phi \mid s)\,d\phi, \tag{2}$$

where $\phi$ parameterizes the distribution over $y$, and $\mathbb{P}_n(\phi \mid s)$ refers to the posterior after making $n$ observations from source $s$, $\mathbb{P}_n(\phi \mid s) = \mathbb{P}(\phi \mid y_1^s, \cdots, y_n^s, s)$.

**Online posterior updates**   Since we are interested in the setting where test data is streaming in, we will update $\phi$ in a recursive fashion per data point. Assuming the test points are independent and identically distributed in location $s$, and imposing the same initial prior on $\phi$ in every $s$, we can derive (see Appendix A.2 for the steps of this derivation),

$$\mathbb{P}_n(\phi \mid s) \propto \mathbb{P}(y_n^s \mid \phi, s)\,\mathbb{P}_{n-1}(\phi \mid s). \tag{3}$$

This corresponds to computing the $n$-th posterior by treating the posterior from the previous $(n-1)$-th step as prior. The computation typically yields the update rule for $\phi$ in closed forms, as long as conjugate distributions are used. The form of this posterior-update equation is reminiscent of the recursive update in *continual learning* (Parisi et al., 2019), with the difference that we have a black-box model, which prevents us from updating neural network weights as developed in that literature. The initial prior is independent of location, $\mathbb{P}_1(\phi \mid s) = \mathbb{P}_1(\phi)$, and can be set to be either uniform or in proportion to $\mathbb{P}(y)$.

**Pseudo-labels**  At test-time, since we do not have true observations, we will approximate $\mathbb{P}_n(\phi \mid s)$ using our model predictions, $\hat{y}_n$ instead of true labels $y_n$ in Eq. 3. The predictions are the ones obtained sequentially from our adjusted predictive distributions (using Eq. 1) at the test points in location $s$. Another alternative is to use the pseudo-labels without performing adjustments first; we evaluate both variants.

**Conditional-shift with $s$**  New locations $s$ also entail a shift in the distribution of $x$, hence our estimate of $\mathbb{P}(y \mid x)$ with a neural network is less reliable. We will show empirically that even with this shift, it is possible to observe benefits in generalization when learning hyper-parameters on validation sets corresponding to different deployment locations.

We summarize the procedure in the algorithm below.

---

**Algorithm 1** Algorithm for online adjustment of predictive distributions

---

**Given:** Black-box model $f_\theta$, validation set with different locations $\{s_v^1, \cdots, s_v^V\}$, initial prior $\mathbb{P}_1(\phi)$.
**Init:** Learn adjustment hyper-parameters on validation set using the same adjustment procedure described below for testing. The specific hyper-parameters are described for classification and regression settings in the following two sections.
**Test-time:** In a new test-time deployment location $s$, perform adjustments online:
$n := 1$
**while** *true* **do**
    Compute $\mathbb{P}(y \mid x_n)$ using trained model $f_\theta(x)$.
    Compute $\mathbb{P}_n(y \mid s)$ using Eq. 2.
    Compute adjusted output $\hat{y}_n^s = \arg\max \mathbb{P}(y \mid x_n, s)$ using Eq. 1.
    Update posterior parameters, $\mathbb{P}_{n+1}(y \mid s)$, using Eq. 3.
    $n := n+1$
**end**

---

## 2.1 Classification

For classification problems, $\mathbb{P}(y \mid x)$ is a categorical distribution modeled by a neural network with parameters $\Theta$. $\mathbb{P}(y)$ is the distribution of training labels, obtained by normalizing class-frequencies in the training set. We use the typical Dirichlet-Categorical forms to model $\mathbb{P}(y \mid s)$, with the parameters $\phi$ in Eq. 2 corresponding to the class-probabilities.

$$\mathbb{P}(\phi \mid s) = \mathrm{Dir}(\alpha^s), \tag{4}$$

$$\mathbb{P}(y \mid \phi) = \mathrm{Cat}(\phi). \tag{5}$$

Given previous $n-1$ predictions, the posterior parameters $\alpha^s$ are updated for the $n$-th point using Eq. 3 as

$$\alpha_n^s(k) = \mathbf{1}[\hat{y}_n^s = k] + \alpha_{n-1}^s(k) = \sum_{i=1}^n \mathbf{1}[\hat{y}_i^s = k] + \alpha_0, \tag{6}$$

where $\hat{y}_i^s$ is the prediction for the $i$-th test point at location $s$ and $\alpha_0$ is the pseudo-count added at initialization (same at all $s$). Using these in Eq. 2 yields the posterior predictive distribution

$$\mathbb{P}_n(y = k \mid s) = \frac{\alpha_n^s(k)}{\sum_{k'=1}^K \alpha_n^s(k')}. \tag{7}$$

In practice, since neural networks are typically trained with a surrogate cross-entropy loss via a SOFTMAX operation, the output distribution is typically not calibrated meaningfully for combination with other probabilistic terms (Guo et al., 2017). With this consideration, we will simply learn hyper-parameters $\tau$ to account for the potential mis-matches in calibration across terms, giving us the following predictive function (in log-space):

$$\arg\max_{y} \ \log \mathbb{P}_{\Theta}(y \mid x) + \tau_s \log \mathbb{P}(y \mid s) - \tau_y \log \mathbb{P}(y). \tag{8}$$

Intuitively, the $\tau$ hyper-parameters also play the role of determining how much of the training prior to "subtract", and how much weight to assign to the pseudo-label based re-adjustment. When these hyper-parameters are learned on a validation set also representing generalized target shift, one can hope to improve at novel test-time deployments.

**Logit adjustment**   Recall that the logits from the neural network $f$ are softmax-ed to produce the predictive distribution and the predicted class for a sample $x$ is given by $\arg\max_k w_k^\top f_\theta(x)$, where $w_k$ is the $k$-th column of the weight matrix for the linear layer for a total of $K$ classes. The modified predictive rule here therefore corresponds to an adjustment of the output logits,

$$\arg\max_{k \in [K]} \ w_k^\top f_\theta(x_n^s) + \tau_s \log \mathbb{P}_{n-1}(y = k \mid s) - \tau_y \log \mathbb{P}(y = k). \tag{9}$$

**Initializing pseudo-counts**   We will consider two possible initialization schemes: (1) we may initialize every element of $\alpha_0$ to be the same, to impose uniform priors for new locations, and (2) we may initialize the pseudo-count as an $\alpha_0$-scaling of the marginal distribution $\mathbb{P}(y)$.

## 2.2   Regression

For regression problems, we use Gaussians to model the target variable,

$$\mathbb{P}(y \mid x) \propto \exp\left( -\frac{\tau_x}{2}\Big(y - f_\theta(x)\Big)^2 \right) \tag{10}$$

$$\mathbb{P}(y) \propto \exp\left( -\frac{\tau_y}{2}\Big(y - \mu_y\Big)^2 \right) \tag{11}$$

where $f_\theta(x)$ is the output of the neural network $f$ with parameters $\theta$, $\tau_x$ is the output precision (inverse of variance, treated as a hyper-parameter); $\mu_y, \tau_y$ are the mean and precision of training-$y$.

The parameters $\phi$ in Eq. 2 are now the mean and precision parameters for $y$ in location $s$. We use the Normal-Gamma distribution to model the posterior, since this is the conjugate distribution for Gaussians with unknown mean and precision (DeGroot, 2004),

$$\mathbb{P}(\mu, \tau \mid s) = \mathcal{N}\Big(\mu \mid \mu_s, \frac{1}{\kappa_s \tau}\Big)\mathrm{Ga}(\tau \mid \alpha_s, \beta_s). \tag{12}$$

Combined with the Gaussian likelihood in Eq. 2, this yields $\mathbb{P}(y \mid s)$ in the form of a Student's $t$-distribution,

$$\mathbb{P}(y \mid s) \propto \left( 1 + \frac{\lambda_s}{2\alpha_s}(y - \mu_s)^2 \right)^{-\frac{2\alpha_s + 1}{2}}, \tag{13}$$

where $2\alpha_s$ is the number of degrees of freedom, and $\lambda_s = \frac{\alpha_s \kappa_s}{\beta_s(\kappa_s + 1)}$. Using these, our predictive function (in log-space) takes the form

$$\arg\min_{y} \ \frac{\tau_x}{2}\Big(y - f_\theta(x)\Big)^2 - \frac{\tau_y}{2}\Big(y - \mu_y\Big)^2 + \frac{2\alpha_s + 1}{2}\log\left(1 + \frac{\lambda_s}{2\alpha_s}(y - \mu_s)^2\right). \tag{14}$$

Setting the derivative *wrt y* to zero yields a cubic equation (see Appendix B.1):

$$m\tau_d y^3 + (m\tau_\mu - 2m\mu_s\tau_d)y^2 + (\tau_d + m\mu_s^2\tau_d - 2m\mu_s\tau_\mu + am)y + (\tau_\mu + m\tau_\mu\mu_s^2 - am\mu_s) = 0, \qquad (15)$$

where

$$a = 2\alpha_s + 1, \qquad (16)$$

$$m = \frac{\lambda_s}{2\alpha_s}, \qquad (17)$$

$$\tau_d = \tau_x - \tau_y, \qquad (18)$$

$$\tau_\mu = \tau_y\mu_y - \tau_x f_\theta(x). \qquad (19)$$

A positive sign of the second derivative of the objective, given by

$$\tau_d + \frac{am(1 - mD^2)}{(1 + mD^2)^2}, \qquad (20)$$

where $D = y - \mu_s$, tells us if a solution is a (local) minima. When we have one real solution with a positive second derivative, we use this; when we have multiple real solutions with positive second derivatives, we pick the one that corresponds to the smallest objective; when we have no real solutions with positive second derivatives, we do not update $\mathbb{P}(y \mid x)$, retaining $f_\theta(x)$ as the solution. Empirically, we find that the condition for no local minima does not arise for optimal choices of hyper-parameters (also see Appendix B.2).

The update equations at the $n$-th step follow from the computation of the posterior using Eq. 3 (see Murphy (2007), for example, for the derivation of these update steps) and are given as:

$$\alpha_s^n = \alpha_s^{n-1} + 1/2, \qquad (21)$$

$$\kappa_s^n = \kappa_s^{n-1} + 1, \qquad (22)$$

$$\mu_s^n = \frac{\kappa_s^{n-1}\mu_s^{n-1} + \hat{y}_s^n}{\kappa_s^{n-1} + 1}, \qquad (23)$$

$$\beta_s^n = \beta_s^{n-1} + \frac{\kappa_s^{n-1}(\hat{y}_s^n - \mu_s^{n-1})^2}{2(\kappa_s^{n-1} + 1)}. \qquad (24)$$

$\kappa$ and $\tau_x$ are chosen using the validation set. Similarly as for classification, we use a calibrating pre-multiplier for the precision $\tau_y$, thus once again giving us three hyper-parameters to tune.

**Initializing priors** Once again, we consider two forms of initial values. In order to place uniform priors over the output range, we will simulate a uniform set of samples over the output range. To place training set priors, we will use the mean and variance from the label distribution in the training set, and in this case $y^{\text{pseudo}} = y^{\text{empirical}}$. The number of initial pseudo-samples are used to set initial values of $\kappa$ and $\alpha$. $\mu = \mathbb{E}[y^{\text{pseudo}}]$ is the mean of the pseudo-samples, and $\beta$ is initialized as $0.5(\kappa - 1)\text{Var}(y^{\text{pseudo}})$ (see Appendix B.3 for details).

## 3 Experiments

We illustrate the applicability of the procedure we described with three proof-of-concept experiments on synthetic datasets, and two challenging, realistic datasets – WILDS-iWILDCAM and WILDS-POVERTYMAP from the WILDS set of OOD generalization benchmarks Koh et al. (2021). We evaluate the method using both PRE as well as POST-adjustment pseudo-labels, as mentioned in Section 2.

### 3.1 Synthetic: MNIST

We split MNIST classes into two subsets: $[0, 1, 2, 5, 9]$ and $[3, 4, 6, 7, 8]$. We use different colors to correspond to different deployment locations. In the training set, we color digits in a particular subset a particular color

Train ($r = 0.99$)

Validation (opposite colors with $r = 0.75$)

Test ($r = -1.0$)

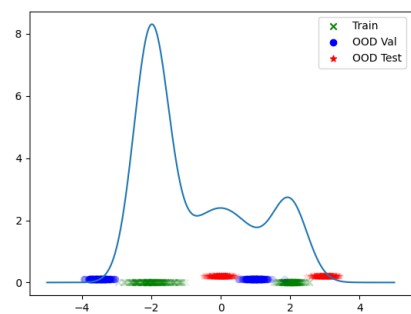

(a) Synthetic variant of the MNIST dataset constructed by using colors to correspond to sources with skewed label-distributions. The colors are flipped for validation and test with different correlation strengths, corresponding to (almost completely) reversing the label-skew at the sources at test-time.

(b) Synthetic MIX-OF-GAUSSIANS data. Differently colored regions along the $x$-axis correspond to training, validation and test samples, with different regions of the same color corresponding to different sources/locations.

Figure 1: Synthetic MNIST and Gaussian datasets.

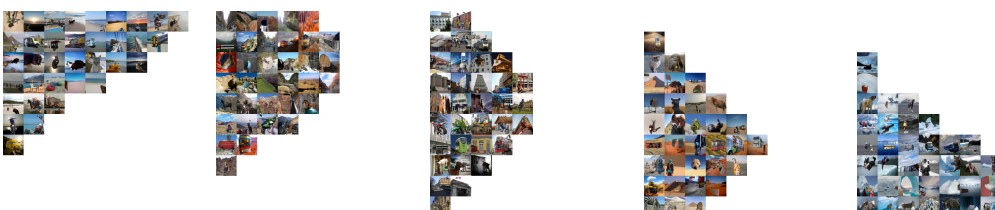

Figure 2: Skewed COCO-on-Places: synthetic dataset constructed by superimposing COCO objects (Lin et al., 2014) on scenes from the Places dataset (Zhou et al., 2017). The 5 columns correspond to 5 sources of data, where the backgrounds correspond to examples of particular scenes, and the skew in number of examples per row correspond to the skew in label distribution we impose. The figure shows samples from the training set; for validation and test sets, different backgrounds are used.

99% of the time. This corresponds to a 99% skew in label-distributions across the two locations. The 1% cross-over would instruct some color-invariance but not strongly enough to completely overcome the bias. The validation set uses opposing colours for the subsets, but with a 75% correlation – this represents a scenario where the class-distributions in different locations change from that in training. Finally, the test set uses completely flipped colors in the two subsets compared to the training set – this implies reversed label-distributions, resulting in poorer baseline performance.

Since the overall class frequencies are balanced in the training set, we drop the $\mathbb{P}(y)$ term, and are only left with $\tau_s$ and $\alpha_0$ to tune through hyper-parameter optimization. This is done using the validation subsets. With a 3-layer CNN trained for 20 epochs to 100% training set accuracy, we find, in Table 1a, that using online adjustments at test-time can lead to marked improvements for the base model in the test set. Validation and test sets were randomly shuffled for 5 trials. (More details in Appendix C.1)

Table 1: On synthetic datasets, base model performance improves significantly when using online adjustments.

| Method | Test accuracy | Method | MSE | Method | Test Accuracy |
|---|---|---|---|---|---|
| Base | $81.00 \pm 0.83$ | Base | $9.17 \pm 2.67$ | Base | $56.09 \pm 0.66$ |
| Online-Pre | $\mathbf{97.50 \pm 0.52}$ | Online-Pre | $4.35 \pm 1.48$ | Online-Pre | $57.92 \pm 1.22$ |
| Online-Post | $95.94 \pm 1.46$ | Online-Post | $\mathbf{4.09 \pm 2.27}$ | Online-Post | $\mathbf{59.07 \pm 0.61}$ |
| (a) Colored MNIST | | (b) Mix-of-Gaussians | | (c) Skewed-COCO-on-Places | |

### 3.2 Synthetic: Mix-of-Gaussians

We create a synthetic regression dataset by constructing a curve from a mixture of Gaussians. We pick regions on the $x$-axis to correspond to training, validation, and test sets, such that every set samples data from two regions each, corresponding to two locations (see Appendix C.2). In Figure 1b, we depict the curve, along with sampling indicators for the different sets and sources. The points have been placed at different heights for clearer visualization of overlaps.

500 points are sampled from the two training regions, and 250 each for the validation and test sets from their assigned regions. We train a 3-layer MLP with BatchNorm and ReLU activations and a mean squared loss for 100 epochs, yielding an in-distribution test mean squared error (MSE) of $\sim 0.15$. In Table 1b we find that online updating reduces the OOD test MSE significantly. Results are aggregates over five trials, with a different random sampling of all data, followed by training and validation each time. Full results and more experimental details are in Appendix C.2).

### 3.3 Synthetic: Skewed-COCO-on-Places

We construct a synthetic dataset by superimposing segmented objects from COCO Lin et al. (2014) on to scenes from the PLACES dataset Zhou et al. (2017), as in Ahmed et al. (2021). The scenes correspond to the notion of a deployment location, albeit with significant intra-location variation. For every such scene-represented source, we use a different class-distribution to simulate source-specific skews in the label distribution. In Fig. 2 the relative number of images per row represent the relative frequency of a particular class at a specific source. There are a total of $\sim 10K$ training images, $\sim 2.5K$ validation images (each for seen and unseen sources), and $\sim 6K$ test images (each for seen and unseen sources).

The validation and test sets are constructed similarly. For in-distribution validation and test sets, the same set of scenes as for training is used (with different instances), and for new-location validation and test sets, different sets of scenes are used. See Appendix C.3 for details about dataset construction.

We train a ResNet-50 for 400 epochs with SGD+Momentum for the underlying model, achieving an in-distribution test accuracy of 75%. Since the overall distribution of classes is close to uniform, we drop the marginal $\mathbb{P}(y)$ term, learning $\tau_s, \alpha_0$ on validation. In Table 1c we find improved performance over the unadjusted base model. Accuracy is aggregated across 5 random orderings of the test set.

### 3.4 WILDS-iWildCam

We use the variant of the IWILDCAM 2020 dataset Beery et al. (2021) curated by the WILDS set of benchmarks for out-of-distribution (OOD) generalization Koh et al. (2021). The data consists of burst images taken at camera traps, triggered by animal motion. The task is to identify the species in the picture, and the locations correspond to the unique camera trap the pictures are from. There are a total of 182 species in this version of the dataset across a total of 323 camera traps. There is significant skew in terms of species distribution across different camera traps, as well as the number of images available for each trap. The training set consists of $\sim 130K$ images from 243 traps; the in-distribution validation set consists of $\sim 7.3K$ images from the same traps as that in the training set but on different dates; the OOD validation set consists of $\sim 15K$ images taken at 32 traps that are different from the ones in the training set; the in-distribution test set consists of $\sim 8.1K$ images taken by the same camera traps as in the training set, but on different dates from both training and validation; finally, the OOD test set consists of $\sim 43K$ images taken at 48 camera traps that arae different from those for all other splits.

Koh et al. (2021) trained ResNet-50 based models along with their curation of this dataset, also evaluating several methods for OOD generalization and releasing all models. We use their models trained with the domain generalization method CORAL Sun & Saenko (2016), since this model has improved performance over the ERM baseline. They released three sets of weights, trained with three random seeds, which result in some variation in performance. Since our goal is to illustrate potential improvements when adjusting outputs from a base model, we evaluate the online scheme for each of the three seeds and report results separately for the seeds, with 5 random orderings each of the test set.

Table 2: WILDS-iWildCam OOD performance – (top) average accuracy, (bottom) Macro F1. Base performance is achieved with the non-adjusted predictions from the underlying neural network model. Online-U refers to the uniform prior scheme, Online-P refers to the scaled-up $\mathbb{P}(y)$ scheme. Online results are aggregates over 5 random orderings of the test set.

| Seed | Base | Online-Pre-U | Online-Pre-P | Online-Post-U | Online-Post-P |
|---|---|---|---|---|---|
| 0 | 70.25 | $71.61 \pm 0.08$ | $71.18 \pm 0.06$ | $\mathbf{71.78 \pm 0.04}$ | $71.64 \pm 0.05$ |
| 1 | 76.79 | $78.79 \pm 0.07$ | $77.71 \pm 0.05$ | $78.65 \pm 0.37$ | $\mathbf{78.81 \pm 0.10}$ |
| 2 | 72.30 | $70.91 \pm 0.07$ | $71.48 \pm 0.10$ | $\mathbf{72.92 \pm 0.07}$ | $70.22 \pm 0.18$ |

| Seed | Base | Online-Pre-U | Online-Pre-P | Online-Post-U | Online-Post-P |
|---|---|---|---|---|---|
| 0 | 32.69 | $33.04 \pm 0.18$ | $31.38 \pm 0.64$ | $33.43 \pm 0.36$ | $\mathbf{34.20 \pm 0.44}$ |
| 1 | 32.91 | $\mathbf{34.44 \pm 0.25}$ | $33.76 \pm 0.31$ | $33.39 \pm 0.15$ | $33.97 \pm 0.51$ |
| 2 | 32.78 | $33.00 \pm 0.15$ | $33.05 \pm 0.15$ | $33.38 \pm 0.32$ | $\mathbf{33.66 \pm 0.07}$ |

Table 3: WILDS-PovertyMap OOD performance – (top) Pearson's correlation coefficient on the entire test set, (bottom) worst-group correlation. Online results are aggregates over 5 random orderings of the test set.

| Fold | Base | Online-Pre-U | Online-Pre-P | Online-Post-U | Online-Post-P |
|---|---|---|---|---|---|
| A | 0.84 | $0.84 \pm 0.00$ | $0.84 \pm 0.00$ | $0.84 \pm 0.00$ | $0.84 \pm 0.00$ |
| B | $\mathbf{0.83}$ | $0.82 \pm 0.00$ | $0.82 \pm 0.00$ | $0.82 \pm 0.00$ | $0.82 \pm 0.00$ |
| C | 0.80 | $\mathbf{0.83 \pm 0.00}$ | $0.80 \pm 0.00$ | $\mathbf{0.83 \pm 0.00}$ | $\mathbf{0.83 \pm 0.00}$ |
| D | 0.77 | $0.77 \pm 0.00$ | $0.77 \pm 0.00$ | $0.77 \pm 0.00$ | $0.77 \pm 0.00$ |
| E | 0.75 | $0.75 \pm 0.00$ | $0.74 \pm 0.00$ | $0.75 \pm 0.00$ | $0.74 \pm 0.00$ |

| Fold | Base | Online-Pre-U | Online-Pre-P | Online-Post-U | Online-Post-P |
|---|---|---|---|---|---|
| A | 0.42 | $\mathbf{0.43 \pm 0.00}$ | $\mathbf{0.43 \pm 0.00}$ | $\mathbf{0.43 \pm 0.00}$ | $0.42 \pm 0.01$ |
| B | $\mathbf{0.52}$ | $0.50 \pm 0.01$ | $0.50 \pm 0.00$ | $0.51 \pm 0.00$ | $0.51 \pm 0.00$ |
| C | 0.42 | $0.56 \pm 0.01$ | $0.56 \pm 0.01$ | $0.57 \pm 0.01$ | $\mathbf{0.58 \pm 0.02}$ |
| D | 0.50 | $0.56 \pm 0.01$ | $\mathbf{0.57 \pm 0.01}$ | $0.56 \pm 0.01$ | $0.56 \pm 0.02$ |
| E | 0.34 | $\mathbf{0.37 \pm 0.00}$ | $\mathbf{0.37 \pm 0.00}$ | $0.36 \pm 0.02$ | $0.36 \pm 0.01$ |

Koh et al. (2021) recommend evaluation with both average accuracy as well as macro-F1 (since some species in the dataset are rare). We perform evaluation with both metrics, but use our own trained models for average accuracy – this is because Koh et al. (2021) trained their models optimizing for macro F1. We similarly trained CORAL-augmented base models optimizing the penalty coefficient and choice of early stopping.

In Table 2, we show results for the base models trained with three seeds for both average accuracy and macro-F1. For evaluating the online-update method, we report mean and standard deviation over five trials, where each trial uses a different random ordering of the OOD test set. $\tau_s, \tau_y, \alpha_0$ are picked on the OOD validation set.

## 3.5 WILDS-PovertyMap

We use the WILDS variant of a *poverty mapping* dataset Yeh et al. (2020). This is a dataset for estimating average household economic conditions in a region through satellite imagery, measured by an asset wealth index computed from survey data. The data comprises 8-channel satellite images with data from 23 African countries. The locations here correspond to different countries.

Due to the smaller size of the dataset, Koh et al. (2021) recommend a five-fold evaluation, where every fold is approximately constructed as follows – 10K images from 13-14 countries in the training set; 1K images from the same countries for in-distribution validation; 1K images from these countries for in-distribution testing; 4K images from 4-5 countries not in the training set for OOD validation; and 4K images from 4-5 countries in neither training nor validation sets.

This regression task is assessed with Pearson's correlation between predicted economic index vs. actual index, as is standard in the literature (Yeh et al., 2020). Following Koh et al. (2021), we split the assessment into overall average as well as worst-group performance, which picks the worst performance across rural/urban subgroups.

As with IWILDCAM, we use the base networks and weights released by Koh et al. (2021), but with our retrained versions for average correlation coefficient (since the validation choices for the released weights were for worst group performance). Once again, we use the CORAL-augmented models, although CORAL does not seem to add significant advantages to ERM for this dataset. We evaluate separately for each fold (which have quite a bit of variance in base performance) with 5 random orderings of the test set.

In Table 3, we find that while there seems generally little to no improvement for average correlation, there are more significant improvements for three of five folds in terms of worst-group performance.

As noted in Koh et al. (2021), a wide range of differences along many dimensions such as infrastructure, agriculture, development, cultural aspects play a role not only in determining wealth-distribution, but also in terms of how the features manifest in different places (for example, different roof types might imply opposite trends for wealthiness in different parts of the world). Such real-world issues imply that validating for OOD performance is bound to be sensitive to problem types and the specific choices of validation sets used to tune hyper-parameters, and the differences that may arise between an OOD validation set and an OOD test set. This issue extends generally to all attempts at OOD generalization.

## 4 Limitations

**Reliability of the underlying predictive model**  Models using pseudo-labels must ensure that the underlying predictive distribution continues to be reliable to a certain extent to benefit from it. This is less clear in settings with label- and conditional-shift. The division by $\mathbb{P}(y)$ in Eq. 1 can be viewed as correcting for skewed label-distributions in the training set Menon et al. (2021). The use of models trained with penalties designed for distributional robustness, as we do for IWILDCAM and POVERTYMAP with CORAL-augmented base models, may help to an extent for robustness to distributional shift. Nevertheless, while we are able to report improvements for most of our settings, we caution that the positive results ought to be taken with a grain of salt. While our results are indicative that output-adjustment methods can be beneficial in online settings for deployment in new environments, it is certainly plausible that the underlying predictive models would cease to be similarly useful under more extreme shifts.

**Validation**  Another aspect affecting test performance is choice of a validation set. In our experiments, we have used official OOD validation splits from WILDS to pick calibration and prior hyper-parameters. In real life, such sets might not exist beforehand and would require targeted curation by the entity (client or provider) performing the location-based adjustment. It is likely that if the validation and test scenarios differ significantly in terms of the extent and nature of label-shift and conditional shift from the training set, performance might be unpredictable. In Appendix D, we report results for IWILDCAM and POVERTYMAP when picking hyper-parameters using the in-distribution validation set and the test set itself (oracle $\tau$s and pseudo-count). We find that while picking hyper-parameters on the in-distribution validation set comes with high variance: we can sometimes observe improvements over the baseline model, but at other times performance can drop significantly. With oracle hyper-parameters, we would not observe any drops below baseline performance, since the learned $\tau$s can simply be shrunk to small values when the test distribution is too challenging to adapt to.

**Unexpected "adversarial" orderings are possible**  In the derivation for Eq. 3, we assumed the test samples in a new location are presented in an IID manner *wrt* the distribution $\mathbb{P}(y \mid s)$. Violating this assumption can sometimes be adversarial: consider a deployment where we initially observe a biased, very high frequency of one particular category at the expense of other equally representative categories in that location. This could skew our predictive distribution significantly so as to continuously predict this category over and over again. As with any AI model when deployed in the wild, it is advisable to incorporate a monitoring scheme to catch unexpected drifts.

## 5 Related work

While the online, black-box setting has not been explored in the context of adapting to label-shift in new deployments with induced conditional shift, there are several closely-related aspects that have been studied in different settings or contexts in the literature.

**Label-shift for classifiers**   Saerens et al. (2002) provides a seminal discussion about adapting the output distribution of a classifier when the test set undergoes label-shift. The update equation mirrors Eq. 1, and relies upon an iterative Expectation Maximization strategy to estimate the new label distribution. This approach presumes access to the entire test set up front, or a sufficiently representative sample.

While more recent works have investigated other ways to estimate label-shift (Lipton et al., 2018; Azizzade-nesheli et al., 2019), they require holding out a random split of IID data to estimate confusion matrices (used for re-training the classifier with importance-weighted ERM), which does not align well with the online, black-box setting we study. Furthermore, it has been recently suggested (Alexandari et al., 2020; Garg et al., 2020) that the simple correction method in Saerens et al. (2002) often outperforms these later methods when combined with calibration. While Alexandari et al. (2020) perform their calibration using a held-out IID validation set for the iterative method, we perform calibration for our online model using a validation set in novel locations since our goal in this paper is to explore the setting where there is an associated conditional shift with change in locations.

There are relatively much fewer precedents in consideration of label-shift in online settings. Very recently, Wu et al. (2021) proposed to continually refine the output distribution by using online gradient methods when the test-time label distribution drifts over time (in the same location). They rely on similar tools as in some of the methods above, inheriting the need for robust estimates of a confusion matrix computed with held-out IID data. We found that applying the method to iWildCam is unsuccessful because the large number of classes and skewed representation in the validation sets result in non-invertible confusion matrices.

**Unsupervised domain adaptation**   While unsupervised domain adaptation (UDA) (Wilson & Cook, 2020) typically assumes no label-shift between source and target domains, Zhang et al. (2013) pointed out that in practical settings, domain shift often tends to be associated with concurrent label-shift, a setting they christened *generalized target-shift* (GeTarS). The GeTarS setup has received limited attention compared to standard UDA. The key ideas in the literature usually involve aligning latent spaces by conditionally matching training and testing points with estimated class weights to account for label-shifts (Tachet des Combes et al., 2020; Kirchmeyer et al., 2022). More recently, with similar practical considerations as ours about data privacy, UDA with black-box source models has been explored by Zhang et al. (2021), in which a target model is trained by querying a source model API, without access to the model internals or training data. Our setup is not amenable to UDA because we presume both an online setting as well as lack of test-time resources to train an AI model tailored to deployment locations.

**Test-time training**   Another emerging line of literature focuses on updating neural network parameters using test data without being able to match training statistics with test statistics, due to the potential lack of access to training data for the same topical reasons – data privacy and large datasets. Some examples include updating the Batch-Norm statistics optimizing for minimum test-time entropy Wang et al. (2021), or using self-supervised pseudo-labels to adapt the feature extraction part of the network Liang et al. (2020). Our setup here can be viewed as a form of test-time training, but in a more constrained setting, with inaccessible model parameters and no resources to replicate an onsite-model by querying the black-box model, e.g. using distillation (Hinton et al., 2015).

**Contextual priors in fine-grained classification**   The use of context-driven priors for fine-grained classification tasks, where context can resolve ambiguity, has been explored in various ways in the literature. Mac Aodha et al. (2019) use a spatio-temporal prior over species distributions in specific locations by inferring these priors from image metadata such as location, time, and photographer information. Similar works Skreta et al. (2020); Chu et al. (2019); de Lutio et al. (2021) have also demonstrated improvements in fine-grained classifications when available or inferable contextual priors are factored into predictive models.

**Out-of-distribution generalization** There has been a recent surge in interest for methods aiming to learn stable or invariant features across different domains/environments/groups Sun & Saenko (2016); Arjovsky et al. (2019); Krueger et al. (2020); Sagawa et al. (2020). Such approaches have been demonstrated to be useful for certain types of distributional shifts, such as with improved minority group robustness Sagawa et al. (2020) and systematic generalization Ahmed et al. (2021).

Our discussion in this paper is complementary to this set of methods in OOD generalization research. One can use an underlying model trained with cross-group penalties that result in improved OOD generalization, and further improve performance by factoring in useful contextual information.

## 6 Conclusion

We discussed the applicability of probabilistic output-adjustments for black-box models in online novel-deployment settings with private training data and inaccessible model weights. We illustrated that approximate Bayesian accounting of label-shift estimated from pseudo-labels can potentially be used to improve performance in situations such as these, which are becoming ever more practically relevant with widespread consumption and larger models.

While invariance to contextual features imposed through cross-environment penalties can be useful in some ways for robustness, there can be situations where specific contextual priors are helpful. In this paper, we demonstrated how use of deployment-location information in associated label-shift scenarios can improve performance. A practitioner can similarly use any other metadata deemed to provide useful contextual information by domain experts in specific applications.

**Broader Impact Statement**

While our experiments show overall positive trends for the most part, in particular for worst-group performance, it should be noted that any method aiming at improvements for novel deployments is never guaranteed to be fail-safe, since too many things can change unexpectedly in the world. In Section 4, we discuss three possible ways approaches such as ours can backfire, when the underlying assumptions are not met. We believe that if such methods are thoughtfully put into practice with guard-rails for detecting failure-modes online if/when they arise (or their use avoided altogether in specific applications, depending on the stakes), the potential positives can outweigh the potential negatives.

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

## A  Derivations

### A.1  Eq. 1

$$\mathbb{P}(y \mid x, s) = \frac{\mathbb{P}(x, s \mid y)\,\mathbb{P}(y)}{\mathbb{P}(x, s)}, \qquad\qquad \text{(Bayes rule)} \qquad (25)$$

$$\propto \mathbb{P}(x, s \mid y)\,\mathbb{P}(y), \qquad\qquad \text{(dropping terms independent of } y) \qquad (26)$$

$$= \mathbb{P}(x \mid y)\,\mathbb{P}(s \mid y)\,\mathbb{P}(y), \qquad\qquad \text{(using assumption } x \perp\!\!\!\perp s \mid y) \qquad (27)$$

$$= \frac{\mathbb{P}(y \mid x)\mathbb{P}(x)}{\mathbb{P}(y)} \frac{\mathbb{P}(y \mid s)\mathbb{P}(s)}{\cancel{\mathbb{P}(y)}} \cancel{\mathbb{P}(y)}, \qquad\qquad \text{(Bayes rule)} \qquad (28)$$

$$\propto \frac{\mathbb{P}(y \mid x)\,\mathbb{P}(y \mid s)}{\mathbb{P}(y)} \qquad\qquad \text{(dropping terms independent of } y) \qquad (29)$$

### A.2  Eq. 3

$$\mathbb{P}_n(\phi \mid s) = \mathbb{P}(\phi \mid y_1^s, \cdots, y_n^s, s), \qquad\qquad (30)$$

$$= \frac{\mathbb{P}(y_1^s, \cdots, y_n^s \mid \phi, s)\,\mathbb{P}(\phi \mid s)}{\mathbb{P}(y_1^s, \cdots, y_n^s)}, \qquad\qquad \text{(Bayes rule)} \qquad (31)$$

$$\propto \mathbb{P}(y_1^s, \cdots, y_n^s \mid \phi, s)\,\mathbb{P}(\phi \mid s), \qquad\qquad \text{(dropping terms independent of } \phi) \qquad (32)$$

$$= \mathbb{P}(y_1^s, \cdots, y_n^s \mid \phi, s)\,\mathbb{P}(\phi), \qquad\qquad \text{(same prior on } \phi \text{ in all } s) \qquad (33)$$

$$= \prod_{i=1}^{n} \mathbb{P}(y_i^s \mid \phi, s)\,\mathbb{P}(\phi) \qquad\qquad \text{(using assumption } y_j^s \perp\!\!\!\perp y_k^s \mid s) \qquad (34)$$

$$= \mathbb{P}(y_n^s \mid \phi, s)\left( \prod_{i=1}^{n-1} \mathbb{P}(y_i^s \mid \phi, s)\,\mathbb{P}(\phi) \right) \qquad\qquad \text{(regrouping terms)} \qquad (35)$$

$$= \mathbb{P}(y_n^s \mid \phi, s)\,\mathbb{P}_{n-1}(\phi \mid s), \qquad\qquad \text{(by definition)} \qquad (36)$$

## B  Regression model

### B.1  Eq. 15

The required distributions are defined as

$$\mathbb{P}(y \mid x) \propto \exp\left( -\frac{\tau_x}{2}\Big(y - f_\theta(x)\Big)^2 \right), \qquad (37)$$

$$\mathbb{P}(y \mid s) \propto \left( 1 + \frac{\lambda_s}{2\alpha_s}(y - \mu_s)^2 \right)^{-\frac{2\alpha_s + 1}{2}}, \qquad (38)$$

$$\mathbb{P}(y) \propto \exp\left( -\frac{\tau_y}{2}\Big(y - \mu_y\Big)^2 \right), \qquad (39)$$

$$(40)$$

which gives us the objective $J = -\log \mathbb{P}(y \mid x, s)$ expressed as

$$J = -\log \mathbb{P}(y \mid x) - \log \mathbb{P}(y \mid s) + \log \mathbb{P}(y) \qquad (41)$$

$$= \frac{\tau_x}{2}\Big(y - f_\theta(x)\Big)^2 - \frac{\tau_y}{2}\Big(y - \mu_y\Big)^2 + \frac{2\alpha_s + 1}{2}\log\left( 1 + \frac{\lambda_s}{2\alpha_s}(y - \mu_s)^2 \right) \qquad (42)$$

The derivative of this objective *wrt* y is

$$\frac{\partial J}{\partial y} = \tau_x(y - \mu_x) - \tau_y(y - \mu_y) + \frac{\frac{2\alpha_s+1}{2} \frac{\lambda_s}{2\alpha_s} . \not{2} . (y - \mu_s)}{1 + \frac{\lambda_s}{2\alpha_s}(y - \mu_s)^2} \tag{43}$$

$$= \tau_x(y - f_\theta(x)) - \tau_y(y - \mu_y) + \frac{(2\alpha_s + 1)\frac{\lambda_s}{2\alpha_s}(y - \mu_s)}{1 + \frac{\lambda_s}{2\alpha_s}(y - \mu_s)^2} \tag{44}$$

$$= \underbrace{(\tau_x - \tau_y)}_{\tau_d} y + \underbrace{(\tau_y\mu_y - \tau_x f_\theta(x))}_{\tau_\mu} + \frac{\overbrace{(2\alpha_s + 1)}^{a} \overbrace{\frac{\lambda_s}{2\alpha_s}}^{m}(y - \mu_s)}{1 + \frac{\lambda_s}{2\alpha_s}(y - \mu_s)^2} \tag{45}$$

$$= \tau_d y + \tau_\mu + \frac{am(y - \mu_s)}{1 + m(y - \mu_s)^2} \tag{46}$$

Setting to zero, we have

$$\left(\tau_d y + \tau_\mu\right)\left(1 + m(y - \mu_s)^2\right) + am(y - \mu_s) = 0 \tag{47}$$

$$\implies \left(\tau_d y + \tau_\mu\right)\left(1 + my^2 + m\mu_s^2 - 2m\mu_s y\right) + am(y - \mu_s) = 0 \tag{48}$$

$$\implies \tau_d y + m\tau_d y^3 + m\mu_s^2\tau_d y - 2m\mu_s\tau_d y^2 + \tau_\mu + m\tau_\mu y^2 + m\tau_\mu\mu_s^2 - 2m\mu_s\tau_\mu y + amy - am\mu_s = 0 \tag{49}$$

$$\implies m\tau_d y^3 + (m\tau_\mu - 2m\mu_s\tau_d)y^2 + (\tau_d + m\mu_s^2\tau_d - 2m\mu_s\tau_\mu + am)y + (\tau_\mu + m\tau_\mu\mu_s^2 - am\mu_s) = 0 \tag{50}$$

which is the equation we shall solve for *y*. We use NUMPYs polynomial solver to find roots. A cubic equation either has one real and a pair of conjugate imaginary roots, or all real roots. We test the real solutions for a positive curvature (implying local minima), and pick the minima resulting in smallest value of the objective *J*.

## B.2   Second derivative test for solutions

The second derivative of *J* is given by

$$\tau_d - \frac{2am^2(y - \mu_s)^2}{(1 + m(y - \mu_s)^2)^2} + \frac{am}{1 + m(y - \mu_s)^2} \tag{51}$$

Writing $y - \mu_s$ as $D$, we have

$$\tau_d + \frac{am}{(1 + mD^2)} - \frac{2am^2 D^2}{(1 + mD^2)^2} = \tau_d + \frac{am}{1 + mD^2}\left(1 - \frac{2mD^2}{1 + mD^2}\right) = \tau_d + \frac{am(1 - mD^2)}{(1 + mD^2)^2} \tag{52}$$

When this expression is positive, we have a local minima.

For the first term to be positive, we require that $\tau_d > 0$, which has a straightforward intuitive interpretation: $\tau_x > \tau_y$, i.e. output precision should be higher than marginal-adjustment precision. This is a reasonable condition which we expect to be fulfilled, since we typically expect to rely more strongly on the underlying predictive model than simply the marginal.

In the second term, $am$ is always non-negative, for a positive pseudo-count. The denominator is always positive. Substituting in expressions for the values after the *n*-th update, we have

$$mD^2 = \frac{\frac{\kappa^n}{\kappa^n+1}(y - \mu^n)^2}{\sum_{t=0}^{n-1} \frac{\kappa^t}{\kappa^t+1}(\hat{y}^{t+1} - \mu^t)^2}. \tag{53}$$

where we have dropped dependence on *s* to reduce clutter. When this term is $\le 1$, we are guaranteed positivity (strictly speaking, $\tau_d$ provides the second term with some room for negative values, but we ignore

it for simplification). This condition implies

$$(y - \mu^n)^2 \leq \frac{\kappa^n + 1}{\kappa^n} \sum_{t=0}^{n-1} \frac{\kappa^t}{\kappa^t + 1} (\hat{y}^{t+1} - \mu^t)^2, \tag{54}$$

which then implies that the following range for $y$ allows local minima

$$\mu^n - \sqrt{\frac{\kappa^n + 1}{\kappa^n} \sum_{t=0}^{n-1} \frac{\kappa^t}{\kappa^t + 1} (\hat{y}^{t+1} - \mu^t)^2} \leq y \leq \mu^n + \sqrt{\frac{\kappa^n + 1}{\kappa^n} \sum_{t=0}^{n-1} \frac{\kappa^t}{\kappa^t + 1} (\hat{y}^{t+1} - \mu^t)^2}. \tag{55}$$

An intuitive interpretation of this condition is that valid updates are allowed within an increasing range as a function of the total observed variances up to the $n$-th test example. In practice, we find that validation tends to pick values for $\tau_x > \tau_y$, and that the case for no-local-minima typically does not arise for the optimal hyper-parameters in our experiments.

### B.3  Initializing priors

For initializing priors, we might endeavour to stay unbiased, since we assume that deployment locations can have significantly different target distributions than we might anticipate from the marginal over the training set. For classification, we built this in by using a uniform pseudo-count for all classes and sources. For regression, we simulate a pseudo-count of uniform samples from the output range.

If we start with a reference prior for the Normal-Gamma distribution with parameter settings

$$\mu = ., \kappa = 0, \alpha = -0.5, \beta = 0, \tag{56}$$

then after observing a $N$ data-points $\{y_1, \cdots, y_N\}, y_i \sim \mathcal{U}[L, H]$ (the uniformly sampled points we will simulate), the resulting posterior is

$$\mu = \frac{1}{N} \sum_{i=1}^{N} y_i, \tag{57}$$

$$\kappa = N, \tag{58}$$

$$\alpha = \frac{N - 1}{2}, \tag{59}$$

$$\beta = \frac{1}{2} \sum_{i=1}^{N} (y_i - \mu)^2. \tag{60}$$

In this view, $\kappa$ corresponds to the pseudo-count (as per the interpretation of the parameters of the Normal-Gamma conjugate prior as in Murphy (2007)). $\alpha$ is defined in terms of $\kappa$. To improve stability, we will set $\mu$ to the middle of the output range rather than actually estimate the mean of our uniform pseudo-samples. Likewise, we will set $\beta$ by estimating its value as a function of $\kappa$ and using the expression for variance of a uniform distribution,

$$\mathbb{E}[\beta] = \frac{1}{2}(\kappa - 1)\mathrm{Var}(y_i) = (\kappa - 1)\frac{(H - L)^2}{24}. \tag{61}$$

For initializing with the scaled-up marginal distribution, we simply plug in the marginal mean and variance into $\mu$ and the formula for $\mathbb{E}[\beta]$ (in place of $\mathrm{Var}(y)$).

## C  Experimental details

### C.1  Synthetic MNIST

The splitting of digits into two sets is performed by observing mis-classification matrices after 200 iterations of training a neural network averaged across a 100 runs – digits are put into opposing sets if they tend to be confused, while also trying to keep the set-sizes balanced.

The network architecture consists of 3 CONV layers with 64, 128 and 256 channels, each followed by MAXPOOL, BATCHNORM, and RELU. After the third layer, we spatially mean-pool activations and use a linear layer to map to the logits. A weight-decay of $5e-4$ is applied on all parameters. Training is conducted for 20 epochs with batches of size 256 where training accuracy saturates to 100%. An initial learning rate of 0.1 is used, which is cut by 5 at the 6-th, 12-th and 16-th epochs.

## C.2 Synthetic Gaussian

The synthetic data for this experiment is generated with the following function

$$y(x) = 10\mathcal{N}(y \mid x; \mu = -2, \sigma = 0.5) + 3\mathcal{N}(y \mid x; \mu = 2, \sigma = 0.5) + 6\mathcal{N}(y \mid x; \mu = 0, \sigma = 1) \tag{62}$$

**Training points:** Training points are sampled from two regions on the $x$-axis, $x \sim \mathcal{N}(-2, 0.4)$ and $x \sim \mathcal{N}(2, 0.2)$, with 250 points each.

**OOD validation points:** OOD validation points are sampled from $\mathcal{N}(-3.5, 0.2)$ and $\mathcal{N}(1, 0.2)$, with 250 points each.

**OOD test points:** OOD test points are sampled from $\mathcal{N}(0, 0.2)$ and $\mathcal{N}(3, 0.2)$, with 250 points each.

For OOD sets, the different sampling distributions correspond to different locations. For different trials, we repeat the whole experiment from scratch, sampling new training, validation, and test sets, and performing validation every time.

The network architecture is a 3 layer MLP with 128 hidden units, with BATCHNORM and RELU after hidden activations. A weight decay of $1e-8$ is applied on all parameters. We train for a 100 epochs with batch-sizes of 100, with SGD + Momentum (0.9), starting with an initial learning rate of 0.01 and scaling it by 0.95 after every epoch.

We include the non-aggregated MSEs to confirm that in spite of the large standard deviations in the aggregated results in Table 1b (due to base performance fluctuations), there are consistent improvements over every base model/data-sampling individually.

| Seed | IID-Base | OOD-Base | OOD-Online |
|------|----------|----------|------------|
| 0 | 0.08 | 11.23 | 4.21 |
| 1 | 0.13 | 12.37 | 6.63 |
| 2 | 0.16 | 6.13 | 2.29 |
| 3 | 0.19 | 9.14 | 1.36 |
| 4 | 0.21 | 7.00 | 5.97 |

## C.3 Synthetic Skewed-COCO-on-Places

We chose the following objects for this synthetic classification task: *bicycle*, *train*, *cat*, *chair*, *horse*, *motorcycle*, *bus*, *dog*, *couch*, and *zebra*; and the following scenes to simulate different sources.

**Training**: beach, canyon, building_facade, desert/sand, iceberg

**OOD validation:** oast_house, orchard, crevasse, ball_pit, viaduct

**OOD test:** water_tower, staircase, waterfall, bamboo_forest, zen_garden

When there are multiple instances of a class in an image, we pick the instance occupying largest area, such that only images with objects occupying at least 10K pixels are retained. All images are resized to $256 \times 256$.

Across the 5 sources, the number of examples for training, validation, and test sets are as follows.

Table 4: Training set

|  | bicycle | train | cat | chair | horse | motorcycle | bus | dog | couch | zebra |
|---|---|---|---|---|---|---|---|---|---|---|
| beach | 669 | 669 | 429 | 176 | 46 | 7 | 0 | 0 | 0 | 0 |
| canyon | 135 | 329 | 513 | 513 | 329 | 135 | 35 | 6 | 0 | 0 |
| building_facade | 5 | 34 | 132 | 322 | 503 | 503 | 322 | 132 | 34 | 5 |
| desert/sand | 0 | 0 | 6 | 35 | 135 | 329 | 513 | 513 | 329 | 135 |
| iceberg | 0 | 0 | 0 | 0 | 7 | 46 | 176 | 429 | 669 | 669 |

Table 5: Validation sets

|  | bicycle | train | cat | chair | horse | motorcycle | bus | dog | couch | zebra |
|---|---|---|---|---|---|---|---|---|---|---|
| beach | 167 | 167 | 107 | 44 | 11 | 1 | 0 | 0 | 0 | 0 |
| canyon | 33 | 82 | 128 | 128 | 82 | 33 | 8 | 1 | 0 | 0 |
| building_facade | 1 | 8 | 33 | 80 | 125 | 125 | 80 | 33 | 8 | 1 |
| desert/sand | 0 | 0 | 1 | 8 | 33 | 82 | 128 | 128 | 82 | 33 |
| iceberg | 0 | 0 | 0 | 0 | 1 | 11 | 44 | 107 | 167 | 167 |

Table 6: Test sets

|  | bicycle | train | cat | chair | horse | motorcycle | bus | dog | couch | zebra |
|---|---|---|---|---|---|---|---|---|---|---|
| beach | 401 | 401 | 257 | 105 | 27 | 4 | 0 | 0 | 0 | 0 |
| canyon | 81 | 197 | 308 | 308 | 197 | 81 | 21 | 3 | 0 | 0 |
| building_facade | 3 | 20 | 79 | 193 | 302 | 302 | 193 | 79 | 20 | 3 |
| desert/sand | 0 | 0 | 3 | 21 | 81 | 197 | 308 | 308 | 197 | 81 |
| iceberg | 0 | 0 | 0 | 0 | 4 | 27 | 105 | 257 | 401 | 401 |

Note that the pattern of label-shift is the same across validation and test subsets. This proof-of-concept experiment is intended as a middle-ground between the COLORED MNIST and WILDS-IWILDCAM experiments, in that the potential of learning hyper-parameters to account for conditional shift is tested while keeping label-shift fixed (recall that the in-distribution test accuracy is 75%, while the OOD test accuracy is markedly lower, at 55%.

We train for 400 epochs with SGD + Momentum (0.9), using batch sizes of 128, with an initial learning rate of 0.1 which is cut by 5 at the 240th, 320th, 360th epochs. An L2 weight decay regulariser is applied on all parameters with a coefficient of $5e-4$. We normalize images with the training set mean and standard deviation per channel, and apply data augmentation of random crops to $224 \times 224$ and random horizontal reflections.

## C.4 Hyperparameters, compute, and code and data licenses.

The hyper-parameters involved are the two calibration terms $\tau_x, \tau_s$ and the pseudo-count term $\alpha$. These were picked via grid-search on validation sets.

V100 GPUs were used to train base models (in cases where we trained our own models), and the online adjustment experiments were performed on an Apple Macbook Air with saved outputs from the models.

We reused code from `https://github.com/p-lambda/wilds`, released under the MIT License. We also used data from MS-COCO, released under the CREATIVE COMMONS ATTRIBUTION 4.0 LICENSE. WILDS-IWILDCAM is under COMMUNITY DATA LICENSE AGREEMENT – PERMISSIVE – V1.0, and the WILDS-POVERTYMAP data is U.S. PUBLIC DOMAIN (LANDSAT/DMSP/VIIRS).

# D    In-distribution and oracle validation

In this section, we report OOD performances when using the in-distribution validation set to pick calibration and pseudo-count hyper-parameters, and when we use the OOD test set itself (oracle hyper-parameters). For online-update methods, all numbers are aggregates over 5 trials corresponding to 5 random orderings of the test set.

Table 7: WIDLS-iWildCam with in-distribution validation. (left) Average accuracy, (right) Macro F1

| Seed | Baseline | Online-U | Online-P | Seed | Baseline | Online-U | Online-P |
|------|----------|----------|----------|------|----------|----------|----------|
| 0 | 70.25 | 72.22 ± 0.48 | **72.58 ± 0.45** | 0 | 32.69 | *32.62 ± 0.18* | *32.61 ± 0.21* |
| 1 | 76.79 | **78.30 ± 0.10** | 78.23 ± 0.10 | 1 | 32.91 | *32.87 ± 0.38* | **33.47 ± 0.32** |
| 2 | 72.30 | **72.72 ± 0.73** | **72.72 ± 0.73** | 2 | 32.78 | *27.90 ± 0.17* | *32.70 ± 0.09* |

Table 8: WILDS-PovertyMap with in-distribution validation. (left) Overall correlation, (right) worst-group correlation

| Fold | Baseline | Online-U | Online-P | Fold | Baseline | Online-U | Online-P |
|------|----------|----------|----------|------|----------|----------|----------|
| A | 0.84 | *0.80 ± 0.00* | *0.72 ± 0.00* | A | 0.42 | **0.43 ± 0.00** | 0.42 ± 0.00 |
| B | 0.83 | *0.82 ± 0.00* | *0.82 ± 0.00* | B | 0.52 | *0.33 ± 0.03* | *0.44 ± 0.02* |
| C | 0.80 | **0.82 ± 0.00** | **0.82 ± 0.00** | C | 0.42 | 0.50 ± 0.01 | **0.52 ± 0.01** |
| D | 0.77 | **0.78 ± 0.01** | 0.75 ± 0.02 | D | 0.50 | *0.46 ± 0.04* | *0.49 ± 0.07* |
| E | 0.75 | *0.72 ± 0.01* | *0.61 ± 0.01* | E | 0.34 | **0.36 ± 0.02** | *0.34 ± 0.01* |

Table 9: WILDS-iWildCam with oracle validation. (left) Average accuracy, (right) Macro F1

| Seed | Baseline | Online-U | Online-P | Seed | Baseline | Online-U | Online-P |
|------|----------|----------|----------|------|----------|----------|----------|
| 0 | 70.25 | 72.23 ± 0.48 | **72.89 ± 0.18** | 0 | 32.69 | 33.58 ± 0.26 | **34.20 ± 0.44** |
| 1 | 76.79 | **79.25 ± 0.07** | 78.85 ± 0.03 | 1 | 32.91 | **35.04 ± 0.93** | 34.15 ± 0.83 |
| 2 | 72.30 | **73.32 ± 0.07** | 73.21 ± 0.08 | 2 | 32.78 | 33.56 ± 0.09 | **33.86 ± 0.17** |

Table 10: WILDS-PovertyMap with oracle validation. (left) Overall correlation, (right) worst-group correlation.

| Fold | Baseline | Online-U | Online-P | Fold | Baseline | Online-U | Online-P |
|------|----------|----------|----------|------|----------|----------|----------|
| A | 0.84 | 0.84 ± 0.00 | 0.84 ± 0.00 | A | 0.42 | **0.45 ± 0.02** | 0.44 ± 0.01 |
| B | 0.83 | 0.83 ± 0.00 | 0.83 ± 0.00 | B | 0.52 | 0.52 ± 0.00 | 0.52 ± 0.00 |
| C | 0.80 | **0.83 ± 0.00** | **0.83 ± 0.00** | C | 0.42 | **0.58 ± 0.02** | **0.58 ± 0.02** |
| D | 0.77 | **0.78 ± 0.00** | **0.78 ± 0.00** | D | 0.50 | **0.57 ± 0.02** | 0.56 ± 0.02 |
| E | 0.75 | 0.75 ± 0.00 | 0.75 ± 0.00 | E | 0.34 | **0.37 ± 0.00** | **0.37 ± 0.00** |

