# OpenReview forum: "Online black-box adaptation to generalized target-shift"
_TMLR — Rejected by TMLR_

### Review · Reviewer_LzRP · 2022-07-19

**Summary Of Contributions:**

This paper studies how to effectively adapt a black box model (both classification and regression) to generalized target-shift problem, which means both conditional shift ($P_{train}(x|y) \neq P_{test}(x|y)$), and label shift ($P_{train}(y) \neq P_{test}(y)$) happens in the deployment time. The paper focuses on the online setting, for which the posterior distribution of the label distribution could be updated regularly. To address the label shift, it utilizes Bayesian tools and pseudo-labels for continuously estimating the label distribution under test domains, which used to re-weigh the black-box predictions. Experiments on three synthetic and two real-world datasets show better performance over the baseline black-box model (w.o. doing any label shift adjustment), which demonstrates the effectiveness of incorporating label-shift to adjustment the output predictions.

**Broader Impact Concerns:**

N/A.

**Requested Changes:**

[1]. The section 2 need to be revised to clearly list the assumptions, such as which distribution is fixed, and which is changing, and discuss how reasonable for these assumptions.

[2]. Some notations are over-loaded, such as $\tau$, it would be great to address them.

[3]. The empirical baselines are weak, and it would be great to incorporate stronger baselines, which also incorporate the test dataset in some way.

[4]. It would be great to show how this method compares with the work in [1], theoretically the paper in [1] gives the sub-optimality regret, and is there some theoretical guarantee for the method in the paper?

[5]. There are some typos in the paper, such as the bottom of Page 4, it should be $\tau_s$ instead of $\tau_x$. Some typos in the appendix also need to be changed, such as Line 27.


**Strengths And Weaknesses:**

Strength:

[1]. The paper studies a very realistic setting, for which label shifts happening between training and deployment time, which makes the black-box model predictions suffers from distributional shifts. Most of the prior works studies this in the offline setting, which might be over-simplistic as the model might be deployed for a long time and the label distribution is rarely to be stationary.

[2]. The paper proposes a simple approach by incorporating posterior distribution of the labels to reweigh the predictions, which in the inner loop also requires the pseudo-label imputations. The paper discusses both the classification and regression tasks, by utilizing a Dirichlet-Categorical model (for label distributions) for classification tasks and Normal-Gamma for regression tasks, and gives explicit updated forms. The methods are easy to implement and gives superior empirical performance.

[3]. The paper is relatively clear and easy to read.


Weakness:

[1]. The paper claims the problem setting is the generalized target shift for which both the conditional shift and label shift are considered, however, the method seems only to correct the label shift, and it seems does not take any step in incorporating the conditional shift. Is there something happen inherently in the method? If not, it would be better to just state that the method only addresses the label shift.

[2]. Some of the assumptions seems not explained in a clear manner, such as in Equation [1], it assumes we have access to a black-box model of $P(y|x)$, typically trained using training data, so should it depends on the training domain $s$ as well? and this makes Equation 1 less convincing.

[3]. For estimating the posterior distribution of $P(\phi|s)$, it uses the pseudo-labels from the predictions that derived from the black-box model, it makes sense however i am wondering as it is the online setting, is it reasonable to incorporate the revealed $y$ labels in the test domain?

[4]. The empirical baselines seems very weak, for example, the work [1] also studies label shift in the online setting, it would be added as a baseline. Though it uses the validation set for the confusion matrix, it could be splitted from the training dataset to estimate it, hence could be served as a reasonable baseline. Also, there are also other baselines from [1] could be borrowed here to demonstrate the effectiveness of the method.

Ref:
[1]. Online Adaptation to Label Distribution Shift, Neurips 2021.

---

> ### Author Response · Authors · 2022-07-22
> **Thanks for the review! Please see our responses.**
>
> Thank you for your quick and thorough review! Here are our comments, and we look forward to the discussion.
>
> Weaknesses:
>
> [1] The goal of our paper is to consider some realistic label-shift scenarios where one encounters conditional-shift along with label-shift. The label-shift literature typically assumes no conditional-shift, and we wanted to explore what happens in realistic datasets such as iWildCam and PovertyMap where both co-occur. Is all hope lost? Or can we still find improvements if we add calibrating hyper-parameters? Our results seem to suggest some promise, and we wanted to bring this potential to the attention of the community. The calibrating hyper-parameters are really what enables an indirect incorporation of conditional-shift, since we tune these with a surrogate OOD validation set (we mention this on Page 3 after Eq. 8). So you are right that there is an implicit and indirect accounting for the conditional-shift, but no explicit method. We are not sure how to express this, do you have any suggestions? We tried to make the problem setting apparent in our abstract where we say: “We explore the use of test-time pseudo-labels \emph{for online label-shift adaptation when deploying black-box models}.”
>
> [2] This is a very valid point, and we discuss it briefly in Section 4. Our reasoning was as follows: even though training domains $s$ is included in $x$, we are not really modelling $P(y|x)$ when we use a neural network; we are really using $P(y | f(x))$ where $f(x)$ is the feature representation. Since the training data has multiple categories per-location, some location-invariance is automatically learned. This is further promoted by the use of CORAL-penalties across $s$ when training the WILDS base models. At the end of the day, we cannot really state with certainty that $s$-information is not in the neural network, but if we look at downstream performance after incorporating our assumptions, it would seem that the overall method is somewhat effective, and simple to implement. Perhaps this is best viewed as a modelling assumption that is imperfect, but useful, and has in fact been used in prior works [Mac-Aodha et al., 2019, Chu et al., 2019] likely for the same reasons.
>
> [3] Using labels is permissible in online training, however we are in the online testing scenario. This typically means true labels are not going to be made available (as discussed in [1]).
>
> [4] In our literature survey, we were unable to find a lot of existing methods that fit the practical scenario we tackle. However, you are right that [1] is applicable in our setting. We did not describe this in detail in the paper, but we had some issues when considering the use of confusion-matrix based methods for our realistic datasets. The first reason is that it wasn’t clear to us how to adapt these methods for regression settings, so we would only be comparing for the classification problems. The other issue is that for such methods, it is necessary to hold our quite a large portion of the training set to estimate a confusion matrix reliably (as we briefly mention in Related Work). When we try to use [1] on the official iWildCam splits, where the number of classes is much higher (182!) and several of the classes are severely under-represented, we ended up with a confusion matrix that is non-invertible. If we use a pseudo-inverse instead, we can proceed, but the performance seems to drop significantly, most likely because of the poor conditioning of the matrix. For example, average accuracy dropped from ~70% down to 33% when using the OGD method, using the automatically computed hyper-parameters like update rate. We are not sure how to present these results, because (1) we operate in a post-hoc setting with trained models that are being deployed, (2) the method isn’t directly applicable for regression problems, and (3) the numbers are quite low, but for understandable reasons (a confusion matrix with a highly imbalanced dataset like iWildCam and in a low-sample regime will be difficult to estimate meaningfully), (4) our goal in this paper is not primarily to show best performance, our goal is just to illustrate that label-shift adaptation can surprisingly work in hard, OOD settings that have not been considered in the label-shift literature, using very cheap updates.
>
> [1]. Online Adaptation to Label Distribution Shift, Neurips 2021.

---

> > ### Author Response · Authors · 2022-07-22
> > **Requested changes**
> >
> > [1] We believe we have listed the assumptions in Section 2; we list them in the first paragraph and the last one. Do you have advice for how to rephrase things?
> >
> > [2] We believe we are not really over-loading $\tau$, since we always add a subscript to specify their use. For example, in classification, $\tau_s$ signifies the scaling on $\log P(y|s)$, and $\tau_y$ signifies scaling applied to the marginal. Are there specific instances you had in mind?
> >
> > [3] Please see our comments above, for weakness [4].
> >
> > [4] Unfortunately, we don’t think we can provide any meaningful theoretical guarantees for our work, precisely because of the concurrent conditional-shift, which makes it hard to say anything solid. OOD generalization is an unpredictable problem in general, but our approach in this submission is to adopt a practical view: we take two hard, medium-to-large-scale realistic datasets exhibiting strong distributional shift problems, and illustrate that with a very simple update rule, one can often surprisingly find improvements. We hope this empirical finding can provide some insight to practitioners in terms of approaches one can take to problems that arise in the real world.
> >
> > [5] The $\tau_x$ referred to at the bottom of page 4 corresponds to precision on the $x$-based predictive distribution (see Eq. 10), and is not associated with the $s$ term. So we believe this is actually not a typo. We’re not sure what you mean by a typo on L27 in the Appendix, could you clarify please?

---

### Review · Reviewer_bXRG · 2022-07-30

**Summary Of Contributions:**

This paper considers the setting of adapting the output of black-box models in an online setting under label-shift. Leveraging Bayesian tools, the authors can estimate the posterior predictive distribution in different deployment locations at test time using the optimal hyper-parameters learned on validation sets. Experimental results show the advantages of their method on both synthetic and realistic datasets.


**Broader Impact Concerns:**

The authors specified three limitations of their method in Section 4. Apart from those, the assumption that every location shares the same initial prior is not realistic in practice. In the classification task, for example, different sources may have different class frequencies or pseudo-count. This scenario is similar to some experimental settings in the paper.

**Requested Changes:**

1) How does the proposed work?

The procedure of the proposal is quite vague. The authors are encouraged to provide an illustration of how to put things together. As far as I understood, there are three main steps. Firstly, a classifier, which is considered a black-box model after this step, is trained on the combination of training datasets from all locations. Then, the validation sets are used to learn hyper-parameters for each location. Finally, these hyper-parameters are used in the test datasets to calculate the posterior.

1.a) How many samples are used in the second and third steps? Do they iterate over all samples in the validation and test sets and if it is, how many times?

1.b) Are the true labels or pseudo labels used in the validation set?

1.c) While the label in the validation and test sets can be IID, the pseudo labels in test sets, which are obtained sequentially from the adjusted predictive distribution, may not satisfy the IID assumption. In addition, from the update formula, the order of test input does matter in the calculation of the posterior predictive distribution. Speaking of which, the authors have already shuffled the test datasets in their experiments.

2) Figure 1a. How is r calculated?

Does r represent the correlation coefficient? Minor: It is a little bit confusing to mix digits between two subsets in the second row considering that the left and right blocks in the first and third rows represent two separate subsets. Alternatively, the authors can put some blue digits in the left block and red digits in the right block.

3) Experimental details.

3. a) What is the best value of hyper-parameters?

3. b) Why does the result of the base model in Table 1c not come with the standard deviation?

4) Baseline.

There is a lack of baseline which is due to the novel setting of the considered problem. However, a simple baseline can be conducted by replacing the proposed estimation of posterior distribution with other estimations. For example, in the classification task, a naive estimation of Equations (6) and (7) can be obtained using pseudo labels without the continual update of Equation (1).
Significance on realistic datasets. The improvement over baseline in realistic datasets in most cases is not significant. This is possibly due to a set of strict assumptions which are discussed in the Limitation section.

5) Minor:
In Section 3.3 it is Fig. 2 instead of Fig. 1


**Strengths And Weaknesses:**

In this paper, the authors consider a novel and interesting online black-box adaptation problem. To the best of my knowledge, this is the first paper that considers such a problem. Experimental results back the efficiency of the proposal even under a high skew level. The experiment settings are sufficient to consider different adaptation scenarios. The authors also conducted ablation studies to compare the results in an ideal setting. The main concerns include the clarity of the method and the lack of experimental details and baselines.

---

> ### Author Response · Authors · 2022-08-04
> **Response to review**
>
> Thanks for the review! Here are our responses to questions.
>
> 1. Yes, your summary is accurate, except that we don’t “learn hyper-parameters for each location”, since we do not know beforehand where we shall deploy our model. We learn the hyper-parameters on a validation set comprising of different locations than at test-time. We added an algorithm box on Page 3 to make the process clearer.
>
>   a) We iterate over all samples in the validation and test sets once in sequence (with different permutations for multiple trials), to mimic the process we will follow at test-time.
>
>   b) The validation scheme uses exactly the same scheme as in test time for learning the hyper-parameters, i.e. we process the validation set sequentially (after random shuffling) using pseudo-labels. Using true labels at validation times (which are available) would result in picking stronger adjustment hyper-parameters, which would not generalize at test-time.
>
>   c) Yes, there is indeed a potential influence of the ordering, and to verify that we are robust to different random orderings, we do shuffle the test sets randomly and average over 5 trials in all experiments. We find we are robust to random reorderings.
>
> 2. $r$ is the probability that a digit in a particular subset appears in a particular deployment location, identified by the color (Section 3.1). The digits are mixed in the second row because we assume that the label-distribution has changed in the locations ($r$ changes from 99% to 75%), so the complementary set of digits are more visible in these locations than in the first row. The left and right blocks do not correspond to digit-color biases, they correspond to different locations associated with a particular “background” color to simulate conditional shfit. Label-shift is simulated by the changing rate of membership of particular digits in the locations.
>
> 3a) Every trial/seed gives rise to a new set of hyper-parameters, and exhaustively reporting them all will take up a lot of space. If we’re not mistaken, the purpose behind this question is to get a broad sense for the ranges and for how the adjustment for P(y|s) trades off with P(y). In general, we observe that $\tau_s$ tends to shift over a range from 0.1 to 10.0. The $\tau_y$ tends to take lower values for average accuracy, around 0.001, and higher values for macro-F1, around 0.5. In regression experiments in PovertyMap, the output precision tends to be higher (around 10.0) for average Pearson’s correlation, implying that adjustments are less preferable, while they are relatively lower (around 5.0) for worst-group correlation.
>
> 3b) We ran experiments with additional trained base models, and have updated the draft with standard deviations for Table 1c.
>
> 4. Thanks for the suggestion! If we understand correctly, you are suggesting that one can either perform the posterior update pre- or post-adjustment. We have added these comparisons to our results. While they are somewhat similar in performance, it appears that pre-updates are generally less reliable than post-updates.
>
> 5. Fixed!
>
> Broader Impact concern: We agree that different locations have different class priors in our experimental settings, and  this is what we are trying to learn online. However, we don’t think there’s any way out of assuming a uniform or training-marginal class prior when starting out. In other words, if we already knew what the location-specific class-skew was, we would have nothing to learn.

---

### Review · Reviewer_ZDEK · 2022-07-31

**Summary Of Contributions:**

The paper presents a Bayesian model for adapting a predictive model in the case of generalized target-shift (GeTarS) in the test time. The generalized target shift means that both P(y) and P(x|y) in the test time could shift from the ones in the training time. The authors consider a black-box setting where we cannot change the underlying parameter of the predictive model, nor do we have access to the training data. In addition, the test data is coming one by one in an online prediction fashion.

The authors started by defining a variable, s, that denotes the different deployment locations, which may be different between train and test phases. The author also assumes the independence of the input data and the deployment location given the label. From these constructions, the authors derive rules to update the posterior distributions after the model saw an additional test sample. The authors then derive two different versions of the models (one for classification setting and one for regression setting), as well as make sure that the distributions involved are conjugate distributions.

Finally, the authors apply the proposed model for both synthetic and real-world datasets, noting the improvement over the base models.

**Broader Impact Concerns:**

I do not have any broader impact concerns for this paper.

**Requested Changes:**

- The authors are suggested to add more motivations, backgrounds, and explanations to the paper in order to make it easier for the reader to understand the contributions of the paper.
- Comparison with previous works as baselines in the experiment section is suggested
- I also suggest the authors to provide illustrations (pictures) and examples on the generalized target shift problems and how the method would address the problems.
- A step-by-step algorithm for explaining the training/adjustment procedure is also suggested.
- Please address the concerns and questions raised in the weaknesses section.

**Strengths And Weaknesses:**

There are a few strong aspects of the paper:
- The problem of target shift is important to tackle as it occurs in real-world machine learning applications.
- The proposed model targets a black-box predictor, which could be useful in practical settings where people already have deployed models and want to adjust the model for target shift.
- The proposed model is applicable to both classification and regression settings.
- The proposed Bayesian technique is suitable for updating distributions in the online learning setting tackled by the paper.

However, the paper also has a few weak parts:
- One of the main weaknesses of the paper is that it does not provide enough background and context for the reader to understand the technique better. The authors start with a brief introduction to the problem. Then, the authors directly jump into method sections without explaining the necessary motivation, background, problem setup, and closely related works; to help understand the method section. Some of the next weaknesses are the results of this.
- The authors aim to address the generalized target-shift (GeTarS) where both test time P(y) and P(x|y) could shift from the training time. However, the authors do not explain this shift in the approach section. Rather, the author introduces the new variable, s, that denotes the deployment location, which may encode the shift. However, the authors do not clearly explain how introducing this variable relates to the generalized target shifts.
- The main assumption in the paper is that the input x and source s are independent given the target y. This seems to be a strong independent assumption, as in the real-world scenario, the deployment location and the value of the input variable may still have relations even given the label value. Could the author provide more explanations and justifications for the assumption since it is the basis of the model construction? The referred papers on this assumption both specifically mentioned geographical settings rather than generic settings.
- There are some approximations in the Bayesian formulation of the method, e.g., the use of pseudo-label in Eq. (3). How does this approximation affect posterior distribution results?
- In the experiments, the authors do not compare the method with any baselines. There are few related works on generalized target shifts. Even though their setting may not be exactly the same as the setting studied in the paper (e.g., non-online learning setting), they can still be used as baselines for comparison. Having baselines comparisons is essential to know what's the benefit of the proposed models compared to the previous works.
- In the experimental results, while the results show that the proposed model provides significant benefits on synthetic datasets, the results for real datasets seem to be mixed. For WildCam dataset, the improvements in the performance over the base model are rather limited, particularly since the base model does not take into account any shifts. For ProvertyMap, in some folds, the proposed model is even worse than the base model. This may raise questions about the effectiveness of the proposed model in real-world datasets.
- [Minor]: Page 12, Eq (27). Shouldn't it be x independent s given y, rather than x independent y given s? Typo?

---

> ### Author Response · Authors · 2022-08-04
> **Response to review**
>
> Thanks for the review!
>
> 1. We have expanded our introduction to provide more motivation.
>
> 2. We do not agree that comparison with methods that operate under different data assumptions and experimental conditions are relevant within the scope of our paper. Our point is specifically to explore and promote a practical yet under-explored area: online black-box settings with realistic distributional shift. Our purpose is to illustrate that in such challenging settings, it can nevertheless be possible to achieve improvements over a base model with cheap adjustments. We believe our set of experiments serves to make this point. Evaluating different methods by changing our experimental settings would change the thrust of our paper. However, at another reviewer’s suggestion, we have added a more relevant comparison to a variant of the adjustment-method, one where we perform the posterior update with the raw pseudo-label before applying adjustments to it.
>
> 3. We believe we have described the problem adequately in the Introduction. The general problem is well-established in the literature; in particular, we heavily reference WILDS, and a reader can easily consult the several pictorial illustrations and detailed analyses of the datasets in that paper.
>
> 4. Thanks for the suggestion — we have added an Algorithm box on Page 3.
>
> 5. Regarding other points:
>
> a. Strong independence assumption: As in the referenced papers where we borrow the independence assumption from, we are primarily concerned with different deployment locations.  An additional line of reasoning is as follows: we are not really modelling $P(y|x)$ when we use a neural network; we are really implicitly using $P(y | f(x))$ where $f$ is the feature representation extracted from the data. Since the training data has multiple categories per-location, some location-invariance is automatically learned. This is further promoted by the use of CORAL-penalties across $s$ when training the WILDS base models. While we cannot be sure that $s$-information has not been picked up, if we look at downstream performance after incorporating our assumptions, it would seem that the overall method is somewhat effective, and simple to implement. Perhaps this is best viewed as a modelling assumption that is imperfect, but useful, and has been used in prior works for the same reasons.
>
> b. Typo on Eq. 27: Thanks for catching this; we’ve fixed it.

---

### Decision · Action_Editors · 2022-09-15

**Recommendation:** Reject

**Comment:**

The paper proposes a post-hoc adjustment technique for settings with label shift. The technique is shown to be particularly useful in settings where one has black-box access to the base model, and seeks to perform online updates.

The reviewers raised a few concerns, summarised below.

_Clarity_. One reviewer suggested that the paper may need to provide more background and context to help frame the method in relation to existing work. From my reading, I tend to agree with this. The current presentation is somewhat terse in parts, which could lead to some confusion:

- the Introduction starts off with a reasonable sumary of generalised target shift, but then starts a discussion of "contextual information" (also referred to as "background" or "non-causal" information) -- it is not clear at this stage what exactly this means. This may be standard material in the literature, but it would be prudent to have the present paper be more self-contained. Further to this, the transition between the first three paragraphs in the Introduction is choppy.

- the Approach section details a setting where there is a source variable controlling the test distribution. It is not sufficiently clear what precisely the assumption is on the train and test distributions Pr(x | y) and Pr(y). It would be useful to spell out Pr(x | y ) and Pr(y) in terms of s.

- when discussing Pr(y | x, s), it ought to be made clear that this is the _ideal_ quantity we want to model, but that s is unknown; the goal of the paper is in identifying a means of easily approximating this quantity. Such details may be immediately apparent to those well-versed in all the background literature, but I believe it would benefit the paper to make such points more explicit.

- it is also not made sufficiently clear that the present paper only attempts to correct for label shift, with conditional shift being accounted for implicitly. There ought to be more foreshadowing of this, e.g., in the Introduction, which in its present form seems to suggest a treatment for general target shifts. The response had some discussion on this that would be useful to include.

_Assumptions_. One critique is that the paper makes the assumption that P(x, s | y) = P(x | y) P(s | y). This assumption certainly will not hold in generality. Its validity depends on the problem setting, but the paper would benefit from discussing more when this might (or might not) be reasonable. The discussion had some arguments related to this, which would be useful to include in the paper.

_Lack of baselines_. One concern was that the paper does not compare itself to any baselines. It was argued that this is owing to the novelty of the problem setting. A suggestion was to compare the method in slightly different settings, which are more amenable to prior work. The response emphasizes that this defeats the claimed focus of the paper, i.e., online settings with black-box models. While I do understand the rationale behind this argument, I tend to agree with the reviewers that it would be insightful to understand how the proposed post-hoc approach would fare in an offline setting. This would help understand the available headroom of improvement on the real-world datasets. Given that the offline setting appears the dominant one in the literature, if the proposed method works well there, that would be of significant interest; if it fares not as well, that would be fine, but at least then one has a sense of whether the online results could potentially be improved by adapting existing algorithms.

The response also details some experiences with a candidate baseline from (Wu et al., '2021). The discussion in the response raised some interesting points regarding the need to have a large validation set, and poor conditioning of the confusion matrix in settings with label skew. It would be useful to incorporate such discussion into the paper: it seems an opportunity to highlight subtle strengths of the proposed method. It also seems natural to consider how the proposed method compares to (Wu et al., '2021) on problems without label skew.

One contemporaneous method for online label shift is:

Bai et al., Adapting to Online Label Shift with Provable Guarantees, arXiV 2207.02121

One method for the related problem of _covariate_ shift in online settings is:

Jang et al., Sequential Covariate Shift Detection, ICML 2022

These papers could be cited. A discussion of the differences in the setting/method of Bai et al. would be interesting.

_Significance of results_. Another concern was that the empirical results are not practically significant on the real-world datasets. It is a bit hard to gauge what scale of improvement would be reasonable, given there are no clear baselines in the online case or corresponding offline model performance. It does appear that on WILDS-PovertyMap, the gains are modest on three folds.

Turning to the TMLR acceptance guidelines:

_Are the claims made in the submission supported by accurate, convincing and clear evidence?_ Per comments above, the clarity of the manuscript could be improved. The paper is fairly measured in its claims about empirical performance, but the lack of baselines make the "convincing" aspect less clear.

_Would at least some individuals in TMLR's audience be interested in knowing the findings of this paper?_ The simplicity of the proposed method is appealing, and the fact that it is useful on some challenging settings could be of interest. It seems likely, however, that readers might want a frame of reference (i.e., baselines) for the results.

The clarity aspect could be improved reasonably easily, but the additional empirical effort on baselines and analyses would require more work, and these would be best served by a fresh round of reviews. Thus, we encourage the authors to work on these changes and submit a revised version of the paper to TMLR for consideration.